# Growth Hormone Therapy in Recurrent Implantation Failure: Stratification by FSH Receptor Polymorphism (Asn680Ser) Reveals Genotype-Specific Benefits

**DOI:** 10.3390/ijms26157367

**Published:** 2025-07-30

**Authors:** Mihai Surcel, Georgiana Nemeti, Iulian Gabriel Goidescu, Romeo Micu, Cristina Zlatescu-Marton, Ariana Anamaria Cordos, Gabriela Caracostea, Ioana Cristina Rotar, Daniel Muresan, Dan Boitor-Borza

**Affiliations:** 1Obstetrics and Gynecology I, Mother and Child Department, “Iuliu Hațieganu” University of Medicine and Pharmacy, 400006 Cluj-Napoca, Romania; mihai.surcel@elearn.umfcluj.ro (M.S.); goidescu.iulian@elearn.umfcluj.ro (I.G.G.); romeo.micu@elearn.umfcluj.ro (R.M.); cristina.rotar@elearn.umfcluj.ro (I.C.R.); muresandaniel01@elearn.umfcluj.ro (D.M.); dan.boitor@elearn.umfcluj.ro (D.B.-B.); 2“Regina Maria” Hospital, 29 Dorobantilor Street, 400117 Cluj-Napoca, Romania; cristina.zlatescu@gmail.com; 3Medical Informatics and Biostatistics, Department XII—Medical Education, Faculty of Medicine, “Iuliu Haţieganu” University of Medicine and Pharmacy, 400006 Cluj-Napoca, Romania; ariana.cordos@elearn.umfcluj.ro; 4Departamentul de Kinetoterapie și Motricitate Specială, Facultatea de Educație Fizică și Sport, Universitatea Babeș-Bolyai, 400347 Cluj-Napoca, Romania; caracostea1@yahoo.com

**Keywords:** recurrent implantation failure, growth hormone, FSH receptor, genotype Asn680Ser, endometrial dysfunction, leukemia inhibitory factor, progesterone

## Abstract

Recurrent implantation failure (RIF) remains a challenging clinical problem. Growth hormone (GH) co-treatment has been explored as an adjunct in poor responders and RIF patients, with inconsistent evidence of benefit. This prospective cohort study assessed the impact of GH supplementation in 91 RIF patients undergoing in vitro fertilization, stratified by FSHR (follicular stimulating hormone receptor) genotype *Asn680Ser* with or without GH supplementation. Patients were stratified by FSHR genotype into homozygous *Ser/Ser* versus *Ser/Asn* or *Asn/Asn* groups. Overall, GH co-treatment conferred modest benefits in the unselected RIF cohort, limited to a higher cumulative live birth rate compared to controls and elevated leukemia inhibitory factor (LIF) levels (*p* < 0.05 both). When stratified by FSHR genotype, the *Ser/Ser* subgroup exhibited markedly better outcomes with GH. These patients showed a higher (0.5 vs. 0.33, *p* = 0.003), produced more embryos (2.88 vs. 1.53, *p* = 0.02), and had a markedly improved cumulative live birth rate—50% with GH versus 13% without—highlighting a clinically meaningful benefit of GH in the *Ser/Ser* subgroup. No significant benefit was observed in *Asn* allele carriers. These findings suggest that FSHR genotyping may help optimize treatment selection in RIF patients by identifying those most likely to benefit from GH supplementation.

## 1. Introduction

Although relatively uncommon, recurrent implantation failure (RIF) represents a significant clinical challenge in assisted reproduction. It remains a poorly defined phenomenon, generally characterized by the repeated failure of embryo transfers deemed viable in women under the age of 40 [1]. This condition often induces profound anxiety and despair in affected couples and contributes to considerable frustration among clinicians. Despite growing scientific interest, progress in refining diagnostic strategies and improving clinical outcomes has been limited [1,2]. Such limited advancement is largely attributed to an incomplete understanding of the underlying pathophysiological mechanisms.

When considering the etiology of RIF, a wide range of contributing factors can be identified, broadly categorized into paternal, embryonic, and maternal components [3,4]. Among these, endometrial dysfunction has increasingly been recognized as a significant contributor in this patient population, particularly when compared to other infertile women [5]. Factors that may impair implantation include parental or embryonic genetic abnormalities, maternal immune dysregulation, hormonal imbalances within the uterine environment, hematologic disorders, alterations in the uterine microbiome, and, notably, embryo–endometrial asynchrony [3,4,6]. Given its multifactorial nature and the complex interplay of contributing elements, RIF is most appropriately characterized as a syndrome rather than a condition with a singular etiology.

An extensive array of strategies for managing RIF has been proposed, including diagnostic tests, pharmacological treatments, surgical interventions, and complementary therapies; yet none have achieved unanimous acceptance within the scientific community [7]. However, growth hormone (GH) stands out as a notable exception, receiving explicit endorsement from the European Society of Human Reproduction and Embryology (ESHRE) guidelines for use in selected cases of RIF [8]. Its therapeutic relevance lies in its multifaceted role, influencing a wide range of reproductive processes—from follicular development and oocyte maturation to enhancing endometrial receptivity [9].

A variety of pathways have been proposed to explain the complex biological effects of GH, involving both direct and indirect mechanisms. The direct actions of GH are evidenced by the presence of its receptors on granulosa and theca cells [10,11], and in the endometrial tissue [12]. This distribution suggests GH can influence both folliculogenesis and endometrial receptivity. Indirectly, GH exerts its impact predominantly through modulation of the insulin-like growth factor (IGF) system, particularly IGF-1, which plays a crucial role in ovarian function and apparently a significant one for the endometrium [11]. Moreover, GH may enhance reproductive outcomes by interacting with key molecular targets, including the follicle-stimulating hormone receptor (FSHR) [13], luteinizing hormone receptor (LHR) [14], bone morphogenetic protein receptor 1B (BMPR1B) [15], leukemia inhibitory factor (LIF) [16], vascular endothelial growth factor (VEGF) [16] and various types of integrins [16]. Among the factors described above, LIF stands out as the most extensively documented marker of endometrial receptivity [17]. As a member of the interleukin-6 cytokine family, it is expressed in both the luminal and glandular epithelium during the early secretory phase, where it interacts with various proteins and signaling molecules crucial for promoting trophoblastic cell motility [17,18]. There is also consistent clinical evidence linking LIF dysregulation to unexplained infertility [19] and RIF [20]. In the same category of indirect implications, some authors suggest that GH might enhance receptivity by improving corpus luteum function, leading to increased progesterone production [16]. Beyond experimental research, substantial clinical evidence supports the beneficial effects of GH, which has been associated with significant improvements in key reproductive outcomes—including a higher number of retrieved oocytes, improved embryo quantity and quality, enhanced endometrial thickness, and increased implantation and live birth rates [21,22]. Despite its promising clinical potential, the routine use of GH continues to face significant hesitation—primarily due to the incomplete understanding of its underlying mechanisms and the absence of clearly defined profiles of patients for whom the treatment would be most advantageous [7].

The follicle-stimulating hormone (FSH) and its receptor system represent a fundamental axis in reproductive biology, playing a pivotal role in orchestrating the final stages of follicular maturation and, ultimately, female fertility [23,24]. Its activation triggers a cascade of biochemical events involving numerous proteins that regulate gene expression and are essential for orchestrating processes such as the cell cycle, cellular proliferation, differentiation, apoptosis, and steroidogenesis [23]. Its complex involvement is further supported by its subtle interactions with several other hormones, notably luteinizing hormone (LH), GH, androgens, and IGF-1 [24,25].

Due to the complexity of the FSH–FSHR regulatory system, patients with unfavorable FSHR genotypes likely exhibit multiple subtle dysfunctions that together shape their clinical profile. Research into FSHR mutations and polymorphisms is particularly significant for patients with suboptimal ovarian responses, as several common variants, including single-nucleotide polymorphisms (SNPs), have been identified in FSHR. Among these, the FSHR *Asn680Ser* polymorphism stands out due to its high prevalence and extensive investigation [26,27,28,29,30,31,32]. It encompasses two linked single nucleotide variants: one in the extracellular domain at position 307 (resulting in either alanine or threonine), and another in the intracellular domain at position 680 (where either asparagine or serine may be present). Three genotypes of the FSHR Asn680Ser polymorphism have been identified: *Ser/Ser*, *Ser/Asn*, and *Asn/Asn*. Among these, the Ser/Ser genotype is widely regarded as the most resistant variant, exhibiting reduced receptor sensitivity to FSH. In contrast, the *Ser/Asn* and *Asn/Asn* genotypes are generally considered functionally similar, showing a more favorable response to FSH stimulation [28,29,30]. Multiple studies have provided compelling evidence that individuals homozygous for the serine allele at position 680 (*Ser/Ser*) typically require significantly higher doses of exogenous FSH for ovarian stimulation, exhibit a reduced number of oocytes retrieved during in vitro fertilization (IVF) cycles [30,31,32,33], and, in certain cohorts, demonstrate lower clinical pregnancy rates compared to carriers of the more responsive genotypes [30,32]. The relationship between recurrent implantation failure (RIF) and *FSHR* polymorphism—particularly the *Asn680Ser (rs6166)* variant—is a topic of emerging interest, but remains under investigation [24,28,29]. Although direct and conclusive evidence linking *FSHR* polymorphism to RIF is currently limited and somewhat inconsistent [31,32], there is a plausible biological rationale supported by indirect evidence, especially in certain patient subgroups. This variability in findings is not unexpected, as the RIF population is heterogeneous, encompassing a wide range of underlying etiologies—some related to folliculogenesis and ovarian function, while others are entirely independent of FSH pathways [2,4].

This study aimed to evaluate the impact of GH administration in patients with RIF, both as a whole group and following stratification by FSHR polymorphism (*Asn680Ser*), to identify potential genotype-specific effects. Furthermore, we sought to investigate potential mechanisms of GH action. In this context, we evaluated a range of follicular parameters—including Follicular Output Rate (FORT), Follicular Output Index (FOI), fertilization rate, and blastulation rate—alongside endometrial markers such as endometrial thickness, LIF levels, and progesterone concentrations on day 5, to explore one of the hypothesized therapeutic pathways of GH. By stratifying patients based on a functional genetic variant (FSHR *Asn680Ser*), our study bridges clinical outcomes with molecular endocrinology, aiming to identify a subset of RIF patients who may particularly benefit from GH, a precision medicine approach in reproductive therapy.

## 2. Results

Based on FSHR genotype and treatment status, patients were classified into four categories: GH—treated *Ser/Ser* (*n* = 16), untreated *Ser/Ser* (*n* = 15), GH—treated *Ser/Asn* + *Asn/Asn* (*n* = 32), and untreated *Ser/Asn* + *Asn/Asn* (*n* = 28).

### 2.1. Baseline Characteristics

The baseline demographic and hormonal parameters were comparable between the GH-treated and untreated groups across genotypes (Table 1). Notably, FSH levels were higher in *Ser/Ser* patients than in those with *Asn* allele, consistent with the known resistance of the *Ser/Ser* genotype (*p* = 0.03 for genotype effect in ANOVA). No other baseline differences have reached significance.

Correlation coefficients reported as Pearson r. Bolded values denote significant correlations (*p* < 0.01).

### 2.2. Overall Impact of GH

When analyzing the entire study cohort, patients treated with GH demonstrated statistically significant improvements in several parameters; however, the clinical relevance of these differences appears modest (Table 2). Significant differences were observed in the number of frozen embryos available (1.0 [0.0–2.0] vs. 1.0 [0.0–1.0], *p* = 0.04), cumulative live birth rate (44% vs. 21%, *p* = 0.04), and LIF levels (27.0 [18.75–39.00] pg./mL vs. 20.0 [12.50–30.00] pg./mL, *p* = 0.02). Additionally, the number of good-quality embryos (3.00 vs. 2.00, *p* = 0.050) and the blastulation rate (0.45 ± 0.18 vs. 0.36 ± 0.11, *p* = 0.050) showed a trend toward improvement, approaching but not reaching statistical significance.

### 2.3. Outcomes by FSHR Genotype

A two-way ANOVA test with Bonferroni correction was employed to evaluate the impact of GH treatment on patients with RIF across various FSHR polymorphism genotypes. Subgroup analysis by FSHR genotype revealed that the clinical benefit of GH treatment was largely confined to *Ser/Ser* (FSHR-resistant) patients

In the *Ser/Ser* genotype subgroup, GH treatment was associated with statistically significant improvements in several outcomes. GH-treated *Se/Ser* had more good-quality embryos (2.88 vs. 1.53, *p* = 0.02) and a higher blastulation rate (0.50 vs. 0.33, *p* = 0.003) than those without GH. There was also a trend toward a higher implantation rate in GH-treated *Ser/Ser* patients (41% vs. 13% *p* = 0.07, Wilcoxon test), although the difference did not reach statistical significance (Table 3).

By contrast, patients with FSHR-sensitive genotypes (*Ser/Asn* + *Asn/Asn*) GH did not have a significantly affected blastulation rate, number of good-quality embryos or any other clinical outcome (Table 3).

### 2.4. Frozen Embryotransfer (FET) Outcomes

In our cohort, 59 patients (64.83%) had at least one embryo suitable for cryopreservation, whereas 32 patients did not yield any embryos for freezing.

Despite a higher number of frozen embryos in GH-treated *Ser/Ser* patients (1.5 vs. 0.0 *p* = 0.01), per transfer outcomes in frozen cycles (implantation rate, clinical pregnancy rate and live birth rates) did not differ significantly between GH and non-GH groups in either genotype category (Table 4). However, when fresh and frozen outcomes were combined the cumulative live birth rate per patient was substantially higher in GH-treated *Ser/Ser* patients than in untreated *Ser/Ser* patients (50% vs. 13%, *p* = 0.05). No such cumulative benefit was observed in the *Ser/Asn* + *Asn/Asn* group (41% vs. 25% *p* > 0.05). Thus, although GH treatment increased embryo availability, implantation and live birth rates per FET remained comparable between groups.

Among endometrial biomarkers, LIF levels were significantly higher in GH-treated *Ser/Ser* patients (27.31 ± 6.8 pg/mL vs. 15.87 ± 5.4 pg/mL, *p* = 0.009). A trend toward statistical significance was observed in progesterone concentrations, although this did not remain significant after Bonferroni correction. No such trend was noted for endometrial thickness across any subgroup (Table 5).

LIF levels showed a moderate positive trend correlated with implantation rate (Pearson *r* = 0.464, *p* < 0.01). LIF was also positively (but weaker) correlated with serum progesterone concentration (*r* ≈ 0.35) and endometrial thickness (*r* ≈ 0.31), although these did not reach statistical significance. Moreover, in a non-parametric analysis, higher LIF and day-5 progesterone were each associated with a greater number of implanted embryos (Spearman’s ρ = 0.235, *p* = 0.034, and ρ = 0.275, *p* = 0.013, respectively), while the correlation between LIF and progesterone levels did not reach statistical significance (*p* = 0.08).

## 3. Discussion

GH administration yielded only modest improvements in reproductive outcomes across the entire RIF cohort, with significant effects primarily observed in cumulative live birth rates and selective enhancements in endometrial parameters, particularly the expression of LIF (Table 2). In contrast, GH had a noticeably stronger effect in patients with the FSHR Ser/Ser genotype. It significantly improved embryo development and appeared to double the clinical pregnancy rate (40% vs. 20%, *p* = 0.07), although this difference did not reach statistical significance (Table 3). In this specific subgroup, GH administration also led to notable increases in endometrial LIF and serum progesterone level factors that were positively correlated with implantation in our study (Table 5). Although implantation rates did not reach statistical significance, a clear trend toward improvement was observed (Table 3). This trend was accompanied by elevated levels of LIF and only a modest increase in progesterone—both of which were positively correlated with implantation—suggesting a potential role for GH in enhancing endometrial receptivity, primarily through an LIF-mediated mechanism (Table 6).

These findings, while preliminary, warrant further investigation. Our study presents specific outcomes that require careful interpretation to fully elucidate their clinical significance. While the overall results of GH administration in the broader RIF cohort may appear modest, they are not entirely unexpected. In contrast to poor responders, whose primary pathology is confined to folliculogenesis, RIF patients constitute a highly heterogeneous population that exhibits dysfunctions across multiple levels [2]. Therefore, the efficacy of targeted treatments may be reduced unless we can accurately identify and address the distinct subgroups within this diverse population.

On the other hand, a substantial and clinically meaningful benefit was observed in patients with presumed FSH signaling dysfunction, specifically those exhibiting the FSHR *Asn680Ser (Ser/Ser)* polymorphic variant. While current evidence emphasizes the significant role of GH through the IGF-I mediator and its direct action via its own receptors [10,11,33], many researchers implicitly acknowledge that the mechanisms currently understood may not be as impactful as once assumed [16]. This suggests that improving embryo or endometrial quality may require exploring additional, previously unrecognized pathways. In this context, the follicle-stimulating hormone system, owing to its remarkably intricate regulatory network and extensive molecular interactions [24], stands out as a pivotal regulator with considerable potential to enhance oocyte quality, while dysfunctions within this pathway may underlie suboptimal ovarian responses [25]. Given the intricate interplay between FSH and GH [24,34], we postulate that in certain RIF patients—particularly those for whom suboptimal ovarian response is partially linked to FSHR dysfunction—adding GH may compensate for that dysfunction by reactivating downstream pathways. Our findings lend credence to this hypothesis, as we observed significantly improved follicular performance (more quality embryos) only in patients with the FSHR (*Ser/Ser*) polymorphism. Conversely, our results indicate that other FSHR variants, namely, *Ser/Asn* and *Asn/Asn*, which possess greater sensitivity to FSH, display only a limited response to GH supplementation. The anticipated beneficial impact of GH on the endometrium was confirmed by a notable trend towards improved implantation rates in fresh cycles, accompanied by elevated levels LIF. However, the precise mechanisms behind this effect remain contentious. Our study emphasizes a significant improvement in the FSH-resistant patient group, suggesting that the corpus luteum may be a critical target for intervention. This group demonstrated a marked increase in day 5 progesterone levels and, crucially, a significant elevation in endometrial LIF levels, underscoring the potential role of GH in enhancing outcomes for this specific population. This group exhibited an increase in day 5 progesterone levels and, more notably, a consistent elevation in endometrial LIF concentrations, suggesting a potential role for GH in enhancing implantation-related endometrial parameters. However, the correlation between LIF and progesterone—used here as a surrogate marker of corpus luteum function—was weaker than anticipated. Accordingly, we refrained from advancing a specific mechanistic pathway—whether mediated by IGF-I, corpus luteum–derived factors, or a direct effect on the endometrium—as our findings only partially supported the original hypothesis. Another noteworthy finding pertains to the reproductive outcomes associated with frozen embryo transfers, where no significant advantage was observed—even within the FSH-resistant group. This suggests that at least part of the increased implantation rate observed in this subgroup during fresh transfers may be attributed to a GH–mediated enhancement of endometrial receptivity. Contrary to our expectations, treatment with GH did not result in a significant improvement in endometrial thickness. This finding may be largely explained by the small number of patients in our cohort presenting with this specific pathology. Collectively, these findings support the hypothesis of an indirect effect of GH on the endometrium, potentially mediated through improved progesterone secretion by the corpus luteum.

When considered within the broader context of GH’s role in IVF, our findings align with the most recent research [21]. Currently, two major and often opposing perspectives exist regarding the use of GH in assisted reproduction. The first supports its administration, highlighting its capacity to enhance ovarian response (particularly in poor responders [35,36,37], its potential to improve embryo quality in patients with RIF [38,39], and its beneficial effect on endometrial receptivity [40,41]. In contrast, the second viewpoint emphasizes significant limitations, including inconsistent clinical outcomes, an incompletely understood mechanism of action, and the lack of standardized protocols for patient selection, dosing, and treatment duration—raising valid concerns about its routine use in clinical practice [42,43,44]. In this regard, a well-designed, recent randomized controlled trial found no significant benefit derived from empiric GH administration in patients anticipated to exhibit a normal ovarian response, underscoring the importance of patient selection when considering GH as an adjuvant therapy [45]. Despite these clear differences, a general agreement is emerging: GH is unlikely to be effective for all patients, but may work best when used selectively, based on specific underlying patient characteristics. Without such refinement, its true therapeutic potential risks being diluted by variable results in heterogeneous patient populations.

A notable discrepancy exists between our findings and those reported by other researchers [28,29,30]. Our data clearly indicate that patients with FSH resistance experience clinically significant dysfunctions that adversely affect both embryo development and endometrial receptivity. In contrast, most studies investigating FSHR polymorphisms primarily focus on variations in time to pregnancy or the FSH doses required for ovarian stimulation, reporting minimal or no substantial impact on overall reproductive outcomes [27,46]. One potential explanation for this divergence lies in our focus on a specifically selected population. Patients in the general population who exhibit an FSH-resistant phenotype may possess multiple compensatory mechanisms, consistent with the complexity of molecular signaling pathways and the plethora of associated regulatory hormones and proteins, enabling them to avoid, compensate for, and ultimately overcome most related dysfunctions [24]. By contrast, among patients with RIF, many individuals may inherently harbor additional ovarian dysfunctions beyond those associated with the FSHR, thereby impairing the compensatory mechanisms that would otherwise support folliculogenesis and oogenesis [5]. In these patients, GH may contribute to the reestablishment of several molecular pathways, either by activating alternative signaling routes —given certain similarities between receptors, such as those involving the mitogen-activated protein kinase (MAPK) pathway [5,12]—or by reinforcing conventional FSH-mediated signaling through direct GH intervention [34].

Emerging evidence suggests that GH can specifically benefit cases of endometrial dysfunction, particularly in thin or non-receptive endometria [12,17]. Notably, several randomized controlled trials (RCTs), along with a recent meta-analysis, have demonstrated that GH administration significantly improves endometrial thickness in infertile women with persistently poor endometrial development (often defined as an endometrial lining < 6 mm or a non-trilaminar endometrial pattern) [17,40,41]. When exploring the underlying mechanisms, most studies—primarily experimental or conducted on animal models—provide evidence for a direct effect of GH on the endometrium through inhibition of the Janus kinase/signal transducer and activator of transcription (*JAK/STAT3*) pathway via regulation of suppressor of Cytokine Signaling 1 (*SOCS1*) in endometrial cells [12,17], or through the modulation of several key proteins, including IGF, LIF, VEGF, and various types of integrins [12,47].

In a broader context, our findings underscore the importance of investigating genetic variants that may lack obvious clinical correlations, possibly due to compensatory mechanisms. While these anomalies may not directly manifest as clinical issues, their association with other pathological factors could contribute to the emergence of specific conditions. Therefore, identifying various polymorphisms within well-defined clinical pathologies, combined with the evaluation of therapeutic responses, will deepen our understanding of the underlying pathophysiology and ultimately enable the delineation of distinct subcategories of RIF patients. From a clinical perspective, testing for the FSHR *Asn680Ser* polymorphism could be a feasible way to identify RIF patients who are more likely to benefit from GH adjuvant therapy. Such stratification is relatively easy to implement (requires a one-time genetic test) and could spare patients unlikely to benefit from GH exposure to unnecessary treatment while concentrating on GH use in those with a plausible biological benefit. This knowledge will ultimately empower clinicians to refine and personalize treatment strategies, improving outcomes for patients facing these complex challenges.

Our study has several limitations, including a small sample size, the absence of randomization, and the use of multiple hypothesis tests without adjusting for multiple comparisons or potential confounders. It is essential to exercise great caution when analyzing the association between progesterone and endometrial receptivity. Compelling evidence suggests that the relationship between luteal progesterone (P4) levels and reproductive outcomes is complex and decidedly non-linear, with both insufficient and excessive luteal P4 levels potentially impairing endometrial receptivity to a significant degree [48]. Thus, an indiscriminate increase in day-5 progesterone should not be assumed a priori to be beneficial, since evidence shows that only patients with initially low progesterone levels benefit from an increase into the optimal physiological range (approximately 150–250 nmol/L). Excessively high mid-luteal P4 can be just as detrimental to receptivity as low P4 [48].

## 4. Materials and Methods

### 4.1. Study Design and Ethical Approval

We conducted a prospective cohort study within the Assisted Reproduction Department of the 1st Obstetrics and Gynecology Clinic in Cluj-Napoca, Romania, between May 2018 and June 2023. This was a non-randomized cohort study, with patient allocation based on consent for GH treatment. Patients who consented to receive GH were assigned to the treatment group, while those who declined served as observational controls. The study was approved by the Ethics Committee of “Iuliu Hațieganu” University of Medicine and Pharmacy (protocol no. 222/10 May 2016) and adhered to the principles of the Declaration of Helsinki. Patients were thoroughly informed about the potential benefits of GH administration and the specifics of the study protocol prior to enrollment. Written informed consent was obtained from all participants before inclusion in the study.

### 4.2. Patient Selection

Consecutive women attending the center for IVF treatment were screened and recruited if they fulfilled the selection criteria. The inclusion criteria were the following: (i) age less than 40 years; (ii) failure to achieve a clinical pregnancy despite transfer of at least four good-quality embryos in at least three IVF cycles (fresh and/or frozen); (iii) a normal uterine cavity confirmed by hysterosalpingography or hysteroscopy. Exclusion criteria: (i) diagnosed with systemic lupus erythematosus, hyper/hypothyroidism, hyperprolactinemia, or having uncontrolled diabetes mellitus; (ii) being treated with androgens or LH or supplementation with antioxidants, such as coenzyme Q10; (iii) presence of untreated hydrosalpinxes on imaging; (iv) diagnosis of moderate to severe endometriosis; (v) presence of abnormal karyotypes in one or both partners; (vi) a baseline day-3 FSH > 15 IU/L; (vii) congenital uterine anomalies; (viii) insufficient information regarding previous embryo transfer cycles.

Furthermore, comprehensive medical history data were systematically collected for each participant at the time of enrollment.

### 4.3. Genetic Testing

Genotyping of the *Asn680Ser (rs6166)* polymorphism in the *FSHR* gene was performed using a TaqMan^®^ SNP Genotyping Assay (Applied Biosystems, Thermo Fisher Scientific, Waltham, MA, USA), following the manufacturer’s protocol. Each reaction was carried out in a final volume of 10 µL, containing 5–10 ng of genomic DNA, TaqMan Genotyping Master Mix, and a validated TaqMan assay mix with allele-specific probes labeled with VIC^®^ and FAM^®^ dyes to discriminate between the *A* (*Asn*) and *G (Ser)* alleles. PCR amplification and allelic discrimination were conducted using the StepOnePlus™ Real-Time PCR System (Applied Biosystems). Genotype calls were automatically generated using StepOne™ Software v2.3, and a subset of samples (10%) was randomly selected and reanalyzed for quality control, yielding 100% concordance.

### 4.4. Study Group and FSHR Stratification

Of the 134 patients deemed eligible for the study, 91 consented to participate. Patients who consented to receive GH treatment were assigned to the treated group, while those who declined were included in the untreated group. Furthermore, all participants underwent genotyping, resulting in the identification of six distinct subgroups based on genotype and treatment status: treated and untreated *Ser/Ser*, *Ser/Asn*, and *Asn/Asn*. However, due to the limited number of patients in the *Asn/Asn* genotype group (17 cases, representing 18.68%), and considering previously reported similarities with the *Ser/Asn* genotype [27,31]—further supported by findings in our own cohort—these groups were consolidated for analysis. Consequently, participants were reclassified into four categories, as also depicted in Figure 1, and listed below.


Group *Ser/Ser* treated.Group *Ser/Ser* untreated.Group *Ser/Asn* + *Asn/Asn* treated.Group *Ser/Asn* + *Asn/Asn* untreated.


### 4.5. Ovarian Stimulation Protocols and Embryo Transfer

Starting doses of gonadotropins (recombinant FSH) were individualized based on patient characteristics, including age, body mass index (BMI), antimüllerian hormone (AMH) levels, and previous IVF experience. For downregulation, patients received either AgGnRH (Triptoreline, 0.1 mg) in a long protocol, or Cetrorelix (Cetrotide, 0.25 mg) or Orgalutran (0.25 mg) in an antagonist protocol.

In addition to the standard GH-treated group (group Ser/Ser-treated and group *Ser/Asn* + *Asn/Asn*-treated), 4 IU daily of recombinant human GH (somatotropin) was administered subcutaneously, commencing on the second day of the IVF cycle and continuing until day 10. The GH-untreated group (group Ser/Ser-untreated and group *Ser/Asn* + *Asn/Asn*-untreated) received only the conventional treatment.

Follicular development was monitored via ultrasound on days 6, 8, 10, and occasionally on day 12 of the IVF cycle. If no follicles measuring ≥12 mm in diameter were detected after 10 days of gonadotropin therapy, the cycle was canceled. When three ovarian follicles ≥17 mm were observed, hCG (Ovitrelle, 250 μg) was administered to trigger final oocyte maturation.

Endometrial thickness was measured in the mid-sagittal plane at the point of maximum thickness of the endometrial stripe in the day of trigger administration.

Oocyte retrieval occurred 34–38 h post-hCG administration. Following retrieval, one to two embryos were transferred on either day three or day five, with any surplus embryos cryopreserved for future use. Luteal phase support was provided through the intravaginal administration of progesterone (Arefam, 200 mg, q.i.d) for 14 days. Pregnancy was confirmed via serum beta-hCG testing 14 days after oocyte retrieval.

Endometrial preparation for FET was achieved using 6–8 mg of oral estradiol valerate daily until endometrial thickness >8 mm was attained. This was followed by the administration of vaginal progesterone (Arefam, 200 mg, q.i.d) and subcutaneous progesterone Prolutex (25 mg s.c. daily) for 14 days, until the confirmation of pregnancy via hCG testing. Hormonal support continued to 12 weeks of gestation if pregnancy was confirmed.

### 4.6. Biological Sample Collection and Analysis

#### 4.6.1. Endometrial LIF Assessment

Endometrial secretions were collected following oocyte retrieval using a sterile soft catheter under ultrasound guidance. Upon reaching the mid-cavity, fluid was aspirated with a 5 mL syringe then aliquoted and stored at −20 °C. LIF concentration was subsequently measured using the Human LIF SimpleStep ELISA Kit (Abcam, Cambridge, UK), following the manufacturer’s instructions.

#### 4.6.2. Serum Progesterone Measurements

To determine serum progesterone concentration, 2 mL of blood was collected on day 6 post-oocyte retrieval. Quantification was performed using a chemiluminescent immunoassay on the Architect Analyzer (Abbott, Abbott Park, IL, USA). The assay exhibited an intra-assay coefficient of variation (CV) of 4.1% and an inter-assay CV of 4.3%. Additional validation was conducted using kits provided by bioMérieux SA (Marcy-l’Étoile, France), which demonstrated an intra-assay CV of 6.6% and inter-assay CV of 8.3%.

### 4.7. Outcomes and Definitions

The primary endpoints of the study encompassed a broad set of ovarian, embryologic, and endometrial parameters relevant to IVF success. These included the following:

FORT—Defined as the ratio between the number of pre-ovulatory follicles (≥17 mm on the day of hCG trigger) and the baseline antral follicle count (AFC), reflecting the efficiency of ovarian response to stimulation;

FOI—Calculated as the ratio between the number of retrieved metaphase II oocytes and the number of mature follicles, serving as a marker of follicular competence and oocyte retrieval efficiency;

Fertilization rate—The proportion of normally fertilized oocytes (2PN—2 pronuclei) out of the total number of mature oocytes;

Blastulation rate—The proportion of embryos that reached the blastocyst stage.

Implantation rate—The percentage of transferred embryos that successfully implanted (visualized as a gestational sac).

Good-quality blastocysts were defined as those with a grade of ≥3 for expansion and grade A or B for both the inner cell mass (ICM) and the trophectoderm (TE), according to the Gardner and Schoolcraft grading system, in line with ESHRE-recommended laboratory practices.

Endometrial thickness—Measured on the day of hCG trigger as an indicator of endometrial receptivity.

LIF levels—Quantified from endometrial secretions as a surrogate marker of molecular receptivity.

Serum progesterone levels—Measured on day 6 post-oocyte retrieval to assess mid-luteal hormonal environment.

Clinical outcome endpoints included the implantation rate, clinical pregnancy rate, and live birth rate (LBR) (evaluated separately for fresh and frozen embryo transfers). The cumulative pregnancy rate (CPR) was defined as the proportion of patients achieving pregnancy across all embryo transfers performed during the study period. Live birth rate (LBR) was calculated per fresh embryo transfer. The cumulative life birth rate (CLBR) per patient was defined as the chance of at least one live birth from all embryo transfers (fresh + associated frozen transfer) steaming from a single ovarian stimulation cycle.

All patients underwent a full transfer cycle, including both fresh and all available frozen embryo transfers, allowing for an accurate calculation of cumulative live birth rate per patient.

### 4.8. Statistical Analysis

All statistical analyses were performed using JASP (Version 0.17.2), a graphical statistical software package. Descriptive statistics were calculated for both clinical and paraclinical variables. Continuous variables with a normal distribution were expressed as mean ± standard deviation (SD), while those not normally distributed were reported as median with interquartile range (IQR).

The following parameters were analyzed: age, duration of infertility, body mass index (BMI), serum levels of FSH, LH, estradiol, AMH, and progesterone; antral follicle count (AFC); total administered FSH dose; number of mature ovarian follicles; number of retrieved oocytes; number of zygotes; fertilization rate; blastulation rate; number of good-quality embryos; implantation rates (fresh and frozen cycles); clinical pregnancy rates (fresh and frozen cycles); abortion rates (fresh and frozen cycles); LIF levels; endometrial thickness; and blood concentrations of testosterone and IGF.

The primary comparison was made between patients who received GH (GH) supplementation and those who did not. The Shapiro–Wilk test was used to assess the normality of distribution for each continuous variable; variables with *p*-values > 0.05 were considered normally distributed.

For normally distributed variables, comparisons between GH-treated and non-treated groups were conducted using the independent samples *t*-test (Student’s *t*-test). For variables not following a normal distribution, the non-parametric Wilcoxon–Mann–Whitney U test was applied. For categorical (binary) outcomes such as implantation rate, miscarriage rate, clinical pregnancy rate, and live birth rates, results were presented as absolute frequencies and percentages. Group comparisons were performed using Fisher’s exact test or Chi-square test, depending on the expected cell counts. Homogeneity of variances was assessed using Levene’s test.

To evaluate the impact of FSHR polymorphisms on clinical and paraclinical outcomes, a two-way analysis of variance (ANOVA) was employed, incorporating two fixed factors: FSHR polymorphism subtype (*Ser/Ser* vs. non *Ser/Ser*) and GH treatment group. An interaction term (FSHR polymorphism × GH treatment) was included to explore potential synergistic or antagonistic effects between genotype and GH administration. When significant main effects or interactions were found, Bonferroni-corrected post-hoc comparisons were performed to identify specific group differences while controlling for type I error. Additionally, subgroup analyses were conducted comparing GH-treated and non-treated patients within unfavorable FSHR genotypes, as well as within favorable FSHR genotypes. All statistical tests were two-tailed, and a *p*-value < 0.05 was considered statistically significant.

## 5. Conclusions

GH supplementation appears to benefit a specific subgroup of RIF patients—those with the FSHR *Ser/Ser* genotype—by enhancing embryonic development and endometrial receptivity in ways that translate into improved pregnancy outcomes. These results support the integration of FSHR genotyping into clinical decision-making, and highlight the need for prospective, genotype-stratified trials to further define the role of GH administration in reproductive medicine.

## Figures and Tables

**Figure 1 ijms-26-07367-f001:**
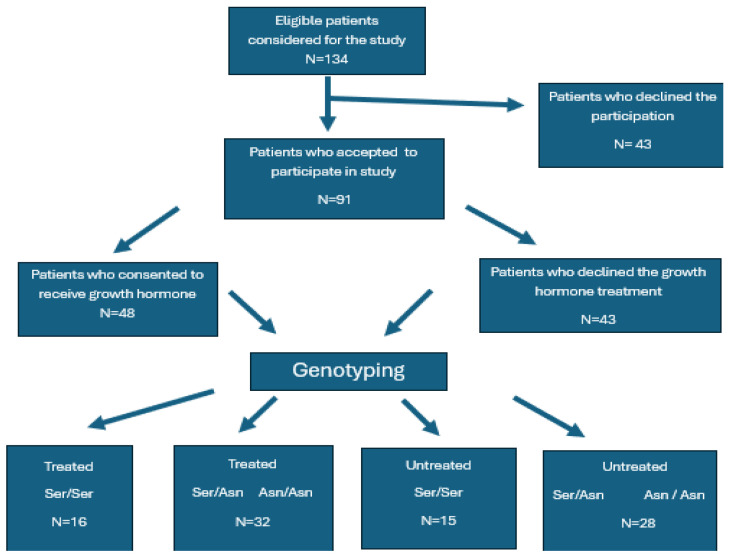
CONSORT diagram of the patient recruitment and allocation process.

**Table 1 ijms-26-07367-t001:** Baseline characteristics by genotype and GH treatment.

Parameter	*Ser/Ser* GH+	*Ser/Ser* GH−	*Ser/Asn* + *Asn/Asn* GH+	*Ser/Asn* + *Asn/Asn* GH−	*p*-Value	Summary Type
Age (years)	37 [34.5–38]	37 [34–37]	38 [34–38]	37 [34–37]	*p* > 0.05	median [IQR]
Infertility duration (years)	5 [4–6.25]	5 [3–7]	5 [4–6]	5 [4–6]	*p* > 0.05	median [IQR]
BMI (kg/m^2^)	24 [22–25]	23 [21.5–24.5]	24 [22–25]	23.5 [22.75–25]	*p* > 0.05	median [IQR]
AMH (ng/mL)	1.58 [0.83–1.86]	1.6 [0.63–1.88]	1.35 [0.73–1.71]	1.45 [0.73–1.97]	*p* > 0.05	median [IQR]
AFC	11.25 ± 2.62	11.73 ± 3.63	11.06 ± 4.33	11.9 ± 3.5	*p* > 0.05	mean ± SD
FSH	9.26 ± 1.84	9.01 ± 2.77	7.12 ± 1.92	7.12 ± 1.55	*p* = 0.03	mean ± SD
LH	6.25 [5.18–6.82]	6.5 [6.25–6.95]	6.05 [5.2–6.82]	6.25 [5.3–7.1]	*p* > 0.05	median [IQR]
Estradiol	53.19 ± 15.95	52.27 ± 17.26	56.75 ± 17.36	48 [38.5–66]	*p* > 0.05	median [IQR]
Previous IVF attempts	3.0 [2.0–3.0]	2.5 [2.0–3.0]	3.0 [2.0–3.0]	3.0 [2.0–3.0]	*p* > 0.05	median [IQR]

BMI—body mass index, AMH—antimullerian hormone, AFC—antral follicle count, FSH—follicle-stimulating hormone, LH—luteinizing hormone, IVF—in vitro fertilization. Note: Values are presented as mean ± SD or %. Data are presented as mean ± SD or median [IQR], depending on distribution.

**Table 2 ijms-26-07367-t002:** Ovarian stimulation, laboratory, and clinical outcomes in GH-treated vs. untreated patients (entire cohort).

Parameter	GH Treated	GH Untreated	*p*-Value
Stimulation days	10.25 ± 1.19	10.14 ± 0.83	*p* > 0.05
Agonist/antagonist protocol (%)	91.67%/8.33%	93.02%/6.98%	*p* > 0.05
Total gonadotropin dose (IU)	2402.08 ± 444.46	2581.4 ± 366.78	*p* > 0.05
Mature follicles	8.71 ± 3.75	8.77 ± 2.77	*p* > 0.05
MII oocytes	7.06 ± 3.45	7.16 ± 2.42	*p* > 0.05
FORT	0.76 ± 0.13	0.72 ± 0.13	*p* > 0.05
FOI (metaphase II oocyte/mature follicle)	0.6 ± 0.16	0.61 ± 0.14	*p* > 0.05
Fertilization rate	0.79 ± 0.12	0.77 ± 0.1	*p* > 0.05
Good-quality embryos	3.00 [1.00–3.00]	2.00 [1.00–2.00]	*p* = 0.05
Blastulation rate	0.40 [0.33–0.50]	0.33 [0.25–0.40]	*p* = 0.05
Implantation rate (fresh)	29%	21%	*p* > 0.05
Clinical pregnancy (fresh)	33%	23%	*p* > 0.05
LIF (pg/mL)	27.0 [18.75–39.00]	20.0 [12.50–30.00]	*p* = 0.02
Progesterone (ng/mL)	79.23 ± 27.34	74.74 ± 27.57	*p* > 0.05
Fresh transfer miscarriage	0 [0–0]	0 [0–0]	*p* > 0.05
Live birth rate (fresh)	25%	16%	*p* > 0.05
Frozen embryos available	1.0 [0.0–2.0]	1.0 [0.0–1.0]	*p* = 0.04
FET implantation rate	16%	22%	*p* > 0.05
FET live birth rate	15%	12%	*p* > 0.05
Cumulative birth rate	44%	21%	*p* = 0.04

GH = growth hormone, IU—international units, MII—metaphase II, FORT—Follicular Output Rate, FOI—Follicular Output Index, FET—frozen embryo transfer. Note: Values are presented as mean ± SD or %. Data are presented as mean ± SD or median [IQR], depending on distribution.

**Table 3 ijms-26-07367-t003:** Ovarian stimulation, laboratory, and clinical outcomes stratified by genotype and GH treatment.

Outcome Parameter	*Ser/Ser* GH+	*Ser/Ser* GH−	*Ser/Asn* + *Asn/Asn* GH+	*Ser/Asn* + *Asn/Asn* GH−	*p*-Value (Resistant)	*p*-Value Resistant (Bonferroni)	*p*-Value (Sensitive)
Stimulation days	10.50 ± 1.21	10.33 ± 0.72	10.13 ± 1.18	10.04 ± 0.88	*p* > 0.05	*p* > 0.05	*p* > 0.05
Total gonadotropin (IU)	2418.08 ± 390.67	2663.63 ± 338.5	2393.75 ± 474.70	2537.5 ± 379.4	*p* > 0.05	*p* > 0.05	*p* > 0.05
Mature follicles	8.31 ± 3.75	8.20 ± 2.76	8.91 ± 4.05	9.07 ± 2.46	*p* > 0.05	*p* > 0.05	*p* > 0.05
MII oocytes	7.13 ± 3.45	6.53 ± 2.72	7.03 ± 3.78	7.50 ± 2.22	*p* > 0.05	*p* > 0.05	*p* > 0.05
I Fort	0.73 ± 0.14	0.63 ± 0.13	0.78 ± 0.10	0.77 ± 0.11	*p* > 0.05	*p* > 0.05	*p* > 0.05
FOI (methtapahase II oocyte/mature follicle)	0.62 ± 0.16	0.50 ± 0.11	0.59 ± 0.14	0.64 ± 0.13	*p* > 0.05	*p* > 0.05	*p* > 0.05
Fertilization rate	0.80 ± 0.10	0.80 ± 0.10	0.78 ± 0.12	0.76 ± 0.17	*p* > 0.05	*p* > 0.05	*p* > 0.05
Blastulation rate	0.50 [0.33–0.58]	0.33 [0.23–0.33]	0.39 [0.33–0.5]	0.39 [0.27–0.5]	*p* = 0.003	*p* = 0.006	*p* > 0.05
Good-quality embryos	2.88 ± 1.66	1.53 ± 0.51	2.22 ± 1.49	1.96 ± 0.59	*p* = 0.02	*p* = 0.04	*p* > 0.05
Implantation rate (%)	41%	13%	29%	25%	*p* = 0.07	*p* > 0.05	*p* > 0.05
Clinical pregnancy rate (%)	40%	20%	31%	32%	*p* > 0.05	*p* > 0.05	*p* > 0.05
Early miscarriage rate (%)	0 [0–0]	0 [0–0]	0 [0–0]	0 [0–0]	*p* > 0.05	*p* > 0.05	*p* > 0.05
Live birth rate (%)	31%	13%	25%	17%	*p* > 0.05	*p* > 0.05	*p* > 0.05

Note: Values are presented as mean ± SD or %. Abbreviations: GH = growth hormone. Data are presented as mean ± SD or median [IQR], depending on distribution, Adjusted *p*-value (Bonferroni).

**Table 4 ijms-26-07367-t004:** FET outcomes by FSHR genotype and GH treatment.

Parameter	*Ser/Ser* GH+	*Ser/Ser* GH−	*Ser/Asn* + *Asn/Asn* GH+	*Ser/Asn* + *Asn/Asn* GH−	*p*-Value (Resistant)	*p*-Value (Sensitive)
No of frozen embryo	1.5 [0.0–3.0]	0.0 [0.0–1.0]	0.0 [0.0–2.0]	0.0 [0.0–1.0]	*p* = 0.01	*p* > 0.05
Frozen implantation rate	19%	28%	16%	20%	*p* > 0.05	*p* > 0.05
Frozen clinical pregnancy rate (%)	31%	13%	19%	18%	*p* > 0.05	*p* > 0.05
Frozen miscarriage rate	0.0 [0.0–1.0]	0.0 [0.0–1.0]	0.0 [0.0–1.0]	0.0 [0.0–1.0]	*p* > 0.05	*p* > 0.05
Frozen live birth/cycle	19%	7%	13%	14%	*p* > 0.05	*p* > 0.05
Total clinical live birth	50%	13%	41%	25%	*p* = 0.05	*p* > 0.05

Note: Values are presented as mean ± SD or %.

**Table 5 ijms-26-07367-t005:** Endometrial and hormonal parameters by GH treatment.

Parameter	*Ser/Ser* GH+	*Ser/Ser* GH−	*Ser/Asn* + *Asn/Asn* GH+	*Ser/Asn* + *Asn/Asn* GH−	*p*-Value (Resistant)	*p*-Value Resistant (Bonferroni)	*p*-Value (Sensitive)
LIF (pg/mL)	27.31 ± 6.8	15.87 ± 5.4	29.13 ± 12.7	25.68 ± 11.5	*p* = 0.009	*p* = 0.01	*p* > 0.05
Progesterone (ng/mL)	87.67 ± 28.43	64.40 ± 34.28	75.28 ± 26.34	73.25 ± 24.71	*p* = 0.05	*p* > 0.05	*p* > 0.05
Endometrial thickness (mm)	8.99 ± 1.64	8.10 ± 1.32	9.89 ± 2.14	9.43 ± 1.79	*p* > 0.05	*p* > 0.05	*p* > 0.05

GH = growth hormone, LIF—leukemia inhibiting factor. Note: Values are presented as mean ± SD or %.

**Table 6 ijms-26-07367-t006:** Correlation matrix: Pearson coefficients among key parameters.

	LIF	Progesterone	Endometrial Thickness	Implantation Rate
LIF	—	0.353	0.308	0.464
Progesterone	0.353	—	0.249	0.154
Endometrial Thickness	0.308	0.249	—	0.238
Implantation Rate	0.464	0.154	0.238	—

GH = growth hormone, LIF—leukemia inhibiting factor. Note: Values are presented as mean ± SD or %.

## Data Availability

Available from the corresponding author upon request.

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
