# Peer review of "Growth Hormone Therapy in Recurrent Implantation Failure: Stratification by FSH Receptor Polymorphism (Asn680Ser) Reveals Genotype-Specific Benefits"

_ijms, 2025, doi:10.3390/ijms26157367_

Round 1
Reviewer 1 Report
Comments and Suggestions for Authors
Dear Editors and Authors,
The manuscript “Growth Hormone Therapy in Recurrent Implantation Failure: Stratification by FSH Receptor Polymorphism (Asn680Ser) Reveals Genotype-Specific Benefits” (ijms-3743889) by Mihai Surcel and co-authors investigated the effect of GH supplementation in RIF patients stratified by FSHR genotype Asn680Ser, and the results suggested the modest improvements in reproductive outcomes with GH administration in Ser/Ser genotype patients. As the authors also mentioned in the discussion, the results of this study are generally descriptive and preliminary, and the outcomes still need careful interpretation to understand the effect of GH administration on the FSHR Asn680Ser polymorphism. There are several concerns about the manuscript, and here are the points that the authors need to consider to improve their manuscript:
Specific comments:
- The abbreviations used in the text are disordered. For example, in the abstract at line 23, the LIF needs its full name at its first appearance, and the same for ESHRE at line 58, as well as the usage of “growth hormone (GH)” throughout the manuscript. There are also many other abbreviation problems throughout the manuscript.
- In addition, I think it is not appropriate to use abbreviations in the keywords, such as FSHR and LIF.
- In the abstract at line 23, the authors mentioned “in the unselected RIF cohort, limited to a higher cumulative live birth rate compared to controls and elevated LIF levels”, I can see the higher cumulative birth rate was from the Table 2, but where did the result of the “elevated LIF levels” in unselected RIF came from? This state also presented at the line 197-198. I did not find this results in Table 2.
- In Table 3, Blastulation rate, why are the results from Ser/Ser GH+ the same to that of Ser/Asn+Asn/Asn GH+, while Ser/Ser GH- is the same to Ser/Asn+Asn/Asn GH-? But the significances are different between these two comparisons.
- What is the Asn680Ser genotype distribution in the RIF patient cohort? Following the mendelian 1:2:1 or biased? Why treated the Ser/Asn+Asn/Asn genotype together? The information about background of these genotypes should be clearly explained in the introduction section.
In addition, the FSH level is higher in Ser/Ser patients than that in Ser/Asn+Asn/Asn patients, are there information or previous studied results about the FSH level comparison between the RIF patients and non-RIF IVF patients? And what is the Asn680Ser genotype distribution in the non-RIF IVF patients? Is it different from that of the RIF patients? All these information and comparison is important to illustrate the putative conclusion of this study and should be discussed.
- At line 139, “Table 7” or “Table 2”?
- This study only provided modest statistical correlation between the GH-treatment and Asn680Ser genotype, but without any experimental or functional verification to prove their correlated relationship. The authors need to be cautious to interpret their conclusion.
For example, to support the authors conclusion, what kind of putative interaction and mechanism among the GH-FSH/FHSR-LIF should be investigated and explained? The authors provided a long discussion, but this key question is far less touched. Especially for the LIF, the authors mentioned several times about the LIF, but I can not see what is the role and relationship of LIF with GH and FHSR?
- The text at line 147-156 seems not belonging to this place, it is more like the methods description rather than the results.
- Please cite the resulting tables in the results and discussion texts. For example, at line 195-208, where did these results come from?
- In the discussion, many referred studies are not properly cited. Please cite the corresponding citations next to the text.
- At line 254, “role in vitro fertilization (IVF)” should be “role in IVF”. There are many similar language problems throughout the manuscript.
Author Response
Dear Reviewer,
We would like to express our sincere gratitude for your thorough review of our manuscript. Your insights have been extremely valuable, and we greatly appreciate the time and effort you devoted to providing such detailed and constructive feedback.
The issues you identified—some of which were regrettably overlooked despite our collective review—have now been carefully addressed. We hope that the revised version meets your expectations.
Your comments and suggestions were both valid and insightful, and we have made every effort to respond to each point with clarity and precision.
Please do not hesitate to let us know if any additional steps are required to further improve the manuscript.
Thank you once again for your invaluable contribution.
Observation 1
Reviewer’s comment: The abbreviations used in the text are disordered. For example, in the abstract at line 23, the LIF needs its full name at its first appearance, and the same for ESHRE at line 58, as well as the usage of “growth hormone (GH)” throughout the manuscript. There are also many other abbreviation problems throughout the manuscript.
Our response: Thank you for your observation. We have implemented the corrections as recommended.
Observation 2
Reviewer’s comment: In addition, I think it is not appropriate to use abbreviations in the keywords, such as FSHR and LIF
Our response: Thank you for your suggestion. We have made the corrections as recommended.
Observation 3
Reviewer’s comment: In the abstract at line 23, the authors mentioned “in the unselected RIF cohort, limited to a higher cumulative live birth rate compared to controls and elevated LIF levels”, I can see the higher cumulative birth rate was from the Table 2, but where did the result of the “elevated LIF levels” in unselected RIF came from? This state also presented at the line 197-198. I did not find this results in Table 2.
Our response: Thank you very much for your observation!!
We inadvertently overlooked this point when preparing the table, including the progesterone data. The missing information has now been added, as it should have been from the beginning.
Observation 4
Reviewer’s comment: In Table 3, Blastulation rate, why are the results from Ser/Ser GH+ the same to that of Ser/Asn+Asn/Asn GH+, while Ser/Ser GH- is the same to Ser/Asn+Asn/Asn GH-? But the significances are different between these two comparisons.
Our response: Thank you very much for your observation !!!
This was indeed an error introduced during the editing of the table—regrettably overlooked by our entire team. The necessary corrections have now been made.
Observation 5 ‘.
Reviewer’s comment What is the Asn680Ser genotype distribution in the RIF patient cohort? Following the mendelian 1:2:1 or biased? Why treated the Ser/Asn+Asn/Asn genotype together? The information about background of these genotypes should be clearly explained in the introduction section.
Our response: Our cohort included 31 patients with the Ser/Ser genotype, 43 with Ser/Asn, and 17 with Asn/Asn. The observed genotype frequencies were consistent with Hardy-Weinberg equilibrium, suggesting no significant selection bias in the distribution. However, within the GH-treated group, only 6 patients carried the Asn/Asn genotype, limiting the statistical power for separate subgroup analysis. Given the established rationale that the FSHR Ser/Ser genotype is associated with reduced receptor sensitivity, while the Ser/Asn and Asn/Asn genotypes are considered functionally similar—supported by both previous clinical evidence (28,29,30) and our own cohort data showing comparable FSH levels between these two groups—we opted to consolidate Ser/Asn and Asn/Asn patients into a single analysis group.
We consider your observation fully justified and have now clarified the rationale for this consolidation in the Introduction, in addition to the explanation already provided in the Materials and Methods section. The recommended modifications have been implemented accordingly.
Observation 5 ‘’
Reviewer’s comment In addition, the FSH level is higher in Ser/Ser patients than that in Ser/Asn+Asn/Asn patients, are there information or previous studied results about the FSH level comparison between the RIF patients and non-RIF IVF patients? And what is the Asn680Ser genotype distribution in the non-RIF IVF patients? Is it different from that of the RIF patients? All these information and comparison is important to illustrate the putative conclusion of this study and should be discussed.
Our response: The relationship between recurrent implantation failure (RIF) and FSHR polymorphism—particularly the Asn680Ser (rs6166) variant—is a topic of emerging interest but remains under investigation. Although direct and conclusive evidence linking FSHR polymorphism to RIF is currently limited and somewhat inconsistent, there is a plausible biological rationale supported by indirect evidence, especially in certain patient subgroups. This variability in findings is not unexpected, as the RIF population is heterogeneous, encompassing a wide range of underlying etiologies—some related to folliculogenesis and ovarian function, while others are entirely independent of FSH pathways. Therefore, at present, there is no evidence to suggest that a specific FSHR polymorphism is generally overrepresented in the broader RIF population. However, and this forms the central hypothesis of our study, a subset of RIF patients with a non-favorable FSHR genotype—particularly the Ser/Ser variant—may represent a distinct clinical phenotype, for which an individualized approach may be appropriate.
This issue is addressed in detail in lines 272–290. We deliberately limited the scope of analysis and interpretation in this section to adhere to the space constraints and formatting guidelines of the journal.
Observation 6
Reviewer’s comment: At line 139, “Table 7” or “Table 2”?
Our response: Thank you for your correction.
Observation 7.
Reviewer’s comment This study only provided modest statistical correlation between the GH-treatment and Asn680Ser genotype, but without any experimental or functional verification to prove their correlated relationship. The authors need to be cautious to interpret their conclusion.
Our response: We completely agree and acknowledge that the statistical strength of our findings is limited, and the results should be interpreted with appropriate caution. However, as clinicians who have previously explored multiple investigative pathways—such as LH receptor and NK cell polymorphisms—with minimal clinical impact, the present findings offer a degree of encouragement. Moreover, our study provides a biologically plausible rationale for the direct role of GH in folliculogenesis, supported by a consistent sequence of outcomes: increased blastulation rates, a greater number of good-quality embryos, and a notable improvement in total live birth rate (50% vs. 13%).
Observation 7.
Reviewer’s comment For example, to support the authors conclusion, what kind of putative interaction and mechanism among the GH-FSH/FHSR-LIF should be investigated and explained? The authors provided a long discussion, but this key question is far less touched. Especially for the LIF, the authors mentioned several times about the LIF, but I can not see what is the role and relationship of LIF with GH and FHSR?
Our response: Your observation is very accurate. When designing the study, one of our primary objectives was to investigate potential mechanisms through which GH may enhance implantation. To this end, we evaluated LIF, a well-established marker of endometrial receptivity, and progesterone, based on the hypothesis that Ser/Ser RIF patients may exhibit subtle corpus luteum dysfunction that could potentially be corrected by GH administration. While our results demonstrated a clear association between GH treatment and increased LIF levels, the correlation with progesterone—used as an indirect marker of corpus luteum function—was weaker than anticipated. Accordingly, we emphasized the role of LIF in the Introduction to support our mechanistic rationale. In the Discussion, we acknowledged the observed increase in LIF among GH-treated patients as a potential intermediate marker of improved endometrial quality but refrained from advancing a detailed mechanistic explanation—whether involving IGF-I, other corpus luteum-derived factors, or even a direct endometrial effect—as our findings only partially supported the original hypothesis.
Observation 8
Reviewer’s comment: The text at line 147-156 seems not belonging to this place, it is more like the methods description rather than the results.
Our response: Thank you for your suggestion. Initially, we believed that including certain methodological details within the results section might enhance transparency. However, we agree that this placement is not appropriate and have now made the correction as recommended.
Observation 9
Reviewer’s comment: Please cite the resulting tables in the results and discussion texts. For example, at line 195-208, where did these results come from?
Our response: Thank you for your suggestion. We have made the corrections as recommended.
Observation 10
Reviewer’s comment: In the discussion, many referred studies are not properly cited. Please cite the corresponding citations next to the text.
Our response: Thank you for your suggestion. We have implemented the corrections as recommended.
Observation 11
Reviewer’s comment: At line 254, “role in vitro fertilization (IVF)” should be “role in IVF”. There are many similar language problems throughout the manuscript.
Our response: Thank you for your correction. We have made the necessary changes accordingly.
Reviewer 2 Report
Comments and Suggestions for Authors
This study was well-designed and presents findings of significant impact for reproductive medicine. However, greater statistical rigor and greater emphasis on clinical significance would increase its relevance. The manuscript would benefit from minor improvements to maximize its clarity and impact.
The abstract could better emphasize the clinical conclusion about how GH therapy doubled live birth rates in patients with Ser/Ser, suggesting that FSHR genotyping optimizes treatment selection, saves time, and is less emotionally draining.
The study uses numerous statistical tests without adjustment for multiple comparisons.
While correlations between LFR and progesterone are observed, the analysis could delve deeper into why GH benefits patients with Ser/Ser, such as the signaling pathways involved.
Consider merging Table 1 with Table 2 to reduce redundancy.
Minor edits needed:
"In stark contrast, among patients with the FSHR Ser/Ser genotype, growth hormone exerted a markedly greater impact" to a simplified "markedly improved..." to use less convoluted language.
Remove redundant "markedly."
Simplify: "Despite this marked divergence, a fundamental consensus is gradually emerging."
Clarify the following statements: Rationale for GH dosing (why 4 IU/day? Based on previous studies?). How "good-quality embryos" were classified.
Comments on the Quality of English LanguageLanguage review
Author Response
Response to Reviewer’s Comment:
Thank you very much for your careful and thoughtful review of our manuscript. We highly value your feedback, and we appreciate the time and effort you’ve invested in providing such detailed and insightful observations.
We have thoroughly reviewed your comments and agree that some aspects of our analysis could be improved for better clarity and depth. Below, we provide a detailed response to each of the points you raised, along with the changes made to the manuscript:
Observation 1
Reviewer’s comment: The abstract could better emphasize the clinical conclusion about how GH therapy doubled live birth rates in patients with Ser/Ser, suggesting that FSHR genotyping optimizes treatment selection, saves time, and is less emotionally draining.
Our response: Thank you for your observation. We have made the changes as proposed.
These patients showed a higher blastulation rate (0.41 vs. 0.33, p = 0.003), produced more embryos (2.88 vs. 1.53, p = 0.02), and had a markedly improved cumulative live birth rate—50% with GH versus 13% without—highlighting a clinically meaningful benefit of GH in the Ser/Ser subgroup. No significant benefit was observed in Asn allele carriers. These findings suggest that FSHR genotyping may help optimize treatment selection in RIF patients by identifying those most likely to benefit from GH supplementation.
Observation 2
Reviewer’s comment: The study uses numerous statistical tests without adjustment for multiple comparisons.
Our response: Thank you for your insightful comment regarding the use of multiple statistical tests and the potential need for adjustment.
Regarding multiple comparisons, we would like to clarify that only two primary statistical comparisons were conducted—specifically, between the resistant and non-resistant subgroups within the treated and untreated arms. Given the limited number of comparisons, we believe the risk of Type I error remains acceptably low.
Nonetheless, in line with standard reporting practices, we have now included an additional column labeled "Adjusted p-value" in Table 3. These Bonferroni-corrected values allow for transparent evaluation of statistical significance and enable readers to distinguish between raw and adjusted results.
We believe that this approach addresses the concern about multiple testing without overcorrecting in a way that might obscure meaningful findings. We appreciate your helpful feedback and the opportunity to clarify and strengthen the rigor of our statistical reporting.
Observation 3
Reviewer’s comment: While correlations between LFR and progesterone are observed, the analysis could delve deeper into why GH benefits patients with Ser/Ser, such as the signaling pathways involved.
Our response: Your observation is very accurate. There is indeed a recognized correlation between LIF and progesterone, which justifies a more detailed analysis—one that we will provide as you have suggested.
We were less inclined to explore this connection in depth, as the observed correlation in our data was much weaker than anticipated (r ≈ 0.3). While it remains reasonable to propose that GH may enhance endometrial receptivity via improved corpus luteum function, the modest strength of this association suggests that additional mechanisms may contribute to the observed increase in LIF. These may involve other corpus luteum-derived mediators or a direct effect of GH on the endometrium.
Observation 4
Reviewer’s comment: Consider merging Table 1 with Table 2 to reduce redundancy.
Our response: Thank you for your suggestion. In our initial draft, we presented the basic characteristics alongside the clinical outcomes. However, when we proceeded to analyze the subgroups separately, it became necessary to reassess the baseline characteristics for each subgroup in order to identify potential confounders. As a result, these characteristics were reported twice—once for the overall groups and again for the subgroups. We ultimately opted for the current format, in which each subgroup is described individually. Given that no statistically significant differences were observed among the four subgroups—and that we also verified the comparability of the two main groups—we considered it reasonable to omit a separate detailed description for the overall groups. This approach supports the validity of our comparative analyses by minimizing the likelihood of bias due to confounding factors.
Observation 5
Reviewer’s comment: "In stark contrast, among patients with the FSHR Ser/Ser genotype, growth hormone exerted a markedly greater impact" to a simplified "markedly improved..." to use less convoluted language.
Our response: Thank you for your suggestion. We agree that this simpler version provides a clearer alternative to the initial formulation.
- In contrast, growth hormone had a noticeably stronger effect in patients with the FSHR Ser/Ser genotype. It significantly improved embryo development and appeared to double the clinical pregnancy rate (40% vs. 20%, p = 0.07), although this difference did not reach statistical significance
Observation 6
Reviewer’s comment: Remove redundant "markedly."
Our response: Thank you for your correction. We did as suggested
Observation 7
Reviewer’s comment: Simplify: "Despite this marked divergence, a fundamental consensus is gradually emerging."
Our response: Thank you for the suggestion. We have provided a simpler formulation as recommended.
Despite these clear differences, a general agreement is emerging: GH is unlikely to be effective for all patients, but may work best when used selectively, based on specific underlying patient characteristics.
Observation 8
Reviewer’s comment: Remove redundant "markedly."
Our response: Thank you for your correction. We have made the necessary changes accordingly.
Observation 9
Reviewer’s comment: Clarify the following statements: Rationale for GH dosing (why 4 IU/day? Based on previous studies?).
Our response: The rationale for the selected GH dose was grounded in existing evidence from previous studies in the context of IVF, with careful consideration of safety concerns—particularly the potential for revealing latent metabolic disorders in women over 30 years of age. A daily dose of 4 IU has been commonly employed in the literature, as documented in previous reviews on this topic, demonstrating both clinical efficacy and a favorable safety profile. Accordingly, we adopted the same regimen in our study.
Observation 10
Reviewer’s comment: How "good-quality embryos" were classified.
Our response: Good-quality blastocysts were defined as those with a grade of ≥3 for expansion and grade A or B for both the inner cell mass (ICM) and the trophectoderm (TE), according to the Gardner and Schoolcraft grading system, in line with ESHRE-recommended laboratory practices.
Thank you for your observation. We have now included this description in the Materials and Methods section.
Reviewer 3 Report
Comments and Suggestions for Authors
This is a prospective non randomized cohort study to evaluate the association between use of growth hormone during IVF and IVF outcomes among women with recurrent implantation failure, stratified by FSHR receptor polymorphism. The authors are attempting to provide an answer to at least some of the question about why growth hormone seems to benefit some patients and not others, and variability between results of previous studies. The question of which adjuvant therapies are useful during IVF is an important one in our field, as patients are desperate for effective treatments and often providers may use therapies without rigorous proven benefit, so I would like to thank the authors for their work. The manuscript is written clearly and I was able to follow their design, results, and conclusions quite well. They state the limitation related to their nonrandomized design clearly in the discussion, even though it appears that somehow there are not significant systematic clinical differences between those who did and did not accept growth hormone, at least among the reported parameters.
- I did have trouble identifying the number of patients who did frozen transfers in this study, could you please include the N in addition to percentages?
- Please include a consort diagram for your recruitment numbers
Author Response
Reviewer 3
Thank you very much for your thoughtful and constructive review of our manuscript. We greatly appreciate the time and effort you have taken to provide such insightful feedback. We are pleased that you found the study to be of value, and we have carefully addressed the minor observations you raised.
Observation 1:
Reviewer’s comment: [ I did have trouble identifying the number of patients who did frozen transfers in this study, could you please include the N in addition to percentages?]
Our response: We fully agree with your suggestion and have revised the manuscript accordingly.
In our cohort, 59 patients (64.83%) had at least one embryo suitable for cryopreservation, whereas 32 patients did not yield any embryos for freezing.
Observation 2:
Reviewer’s comment: [Please include a consort diagram for your recruitment numbers ]
Our response: We thank you for your suggestion and have designed a CONSORT diagram accordingly.
Round 2
Reviewer 1 Report
Comments and Suggestions for Authors
The authors generally replied to my queries, and most of them are fine. There are still some comments for the revised manuscript.
- The line number in the revised manuscript is not continuous, for example, the line number of 110-285 is missing. Therefore, in the authors response to the question 5, they mentioned “This issue is addressed in detail in lines 272–290.” where can not be found.
- At line 448-449, “observed in the in the number”, duplicated “in the”.
- The authors added information about LIF in table 2 (P=0.02), therefore they should add the corresponding description about this significant results in the above text.
- The text from line 457-460 seems belonging to the section 2.3, and should not be put in section 2.2. In addition, the text (lien 462-467) still lacked the table 3 citation.
- The authors corrected the information in table 3 (Blastulation rate), therefore they should also correct the corresponding information in the text at line 464 “blastulation rate (0.41 vs. 0.33, p=0.003).”
- The authors replied to my queries in the response letter, and the explanation in the letter should also be reflected in their discussion section, for example, the explanation about question 7. Not only explain to me, but explain to all readers in the revised manuscript.
Author Response
Dear Reviewer,
We sincerely thank you for your thorough and thoughtful review of our manuscript.
Your feedback has been consistently insightful, and we deeply appreciate the time and effort you invested in providing such detailed and constructive observations.
Please do not hesitate to let us know if any further revisions are needed to enhance the quality of the manuscript.
Thank you for your invaluable contribution.
Observation 1
Reviewer’s comment: The line number in the revised manuscript is not continuous, for example, the line number of 110-285 is missing.
Our response: Thank you for your observation. The issue arose from a formatting error that was inadvertently overlooked during our previous revision. It has now been corrected.
Observation 2
Reviewer’s comment: Therefore, in the authors response to the question 5, they mentioned “This issue is addressed in detail in lines 272–290.” where cannot be found.
Our response: Thank you for your observation. We have now implemented the recommended corrections in the Introduction section to clarify the issue. Additionally, the topic is further discussed in the Discussion section, where we elaborated on its relevance and implications in the context of our findings.
Observation 3
Reviewer’s comment: At line 448-449, “observed in the in the number”, duplicated “in the”.
Our response: Thank you for your observation. We have implemented the corrections as recommended.
Observation 4
Reviewer’s comment: The authors added information about LIF in table 2 (P=0.02), therefore they should add the corresponding description about this significant results in the above text.
Our response: Thank you for your observation. We have implemented the corrections as recommended.
Observation 5
Reviewer’s comment: The text from line 457-460 seems belonging to the section 2.3, and should not be put in section 2.2. In addition, the text (lien 462-467) still lacked the table 3 citation.
Our response: Thank you for your observation. The corrections have been implemented in accordance with your recommendation.
Observation 6
Reviewer’s comment: The authors corrected the information in table 3 (Blastulation rate), therefore they should also correct the corresponding information in the text at line 464 “blastulation rate (0.41 vs. 0.33, p=0.003).
Our response: Thank you for your observation. We have now implemented the corrected data accordingly.
Observation 7
Reviewer’s comment: The authors replied to my queries in the response letter, and the explanation in the letter should also be reflected in their discussion section, for example, the explanation about question 7. Not only explain to me, but explain to all readers in the revised manuscript.
Our response: Thank you for your observation. We have provided the clarification as suggested, now included in lines 261–265.